# Error Analysis Method of Geometrically Incomplete Similarity of End-Plate Connection Based on Linear Regression

**Dongzhuo Zhao** [1] , **Zhan Wang** [1,2,*], **Jianrong Pan** [1,2] **and Peng Wang** [1]

[1] School of Civil Engineering and Transportation, South China University of Technology, Guangzhou 510641, China; zdcom2@126.com (D.Z.); ctjrpan@scut.edu.cn (J.P.); wangp@scut.edu.cn (P.W.)
[2] State Key Laboratory of Subtropical Building Science, South China University of Technology, Guangzhou 510641, China
\* Correspondence: wangzhan@scut.edu.cn

**Abstract:** Due to the limitations of processing errors, test conditions and other factors, geometric similarity errors in scale tests of steel structure joints are difficult to avoid, but the research on this error is little known. Based on the similarity theory and the basic idea of the component method, this paper deduces the similar macro conditions of beam–column end-plate connections and derives the main influencing factors of geometric similarity of these types of structures. Aiming at the factor of the thickness of the end-plate, the formation mechanism of the geometrically incomplete similarity error of this type of node was studied. Through the establishment of accurate finite element models for parameterized analysis, the influence of end plate thickness on incomplete similarity error is analyzed. Based on this model and linear regression analysis methods, the prediction formulas of geometric incomplete similarity errors of beam–column end-plate connections have been established, which can significantly reduce similar errors due to end plate thickness. This article aims to propose a method for simulating the distribution of incompletely similar errors and provide a reference for the research of similar problems.

**Keywords:** structural model test; incomplete geometric similarity; beam–column end-plate connection; sensitivity analysis; regression analysis

---

## 1. Introduction

Beam–column end-plate connection is an important connection form for prefabricated steel structures. For many years, scholars have conducted significant experimental research on this connection form and obtained many interesting results [1–3]. The scale model test is an important type of node test. Under conditions of similar error control, good empirical results can be obtained. At the same time, compared with full-scale model tests, it can save considerable test expenses, site occupation, and time costs. However, due to the lack of effective communication and unified organization, each researcher only focuses on his own research content and rarely analyzes the similar errors of the trial itself, which results in unevenness in the trial results and a waste of experimental resources. Due to this situation, it is urgent to analyze incomplete similar factors in the scale model test for the beam–column end-plate connections and solve the bottlenecks caused by the scale model test.

In view of the performance characteristics of common semi-rigid connections of steel structures, including beam–column end-plate connections, scholars from various countries have performed many experiments and theoretical studies. Lu Linfeng et al. completed the failure tests of six full-scale beam–column connections under low-cycle repeated loads and compared and analyzed the

deformation distribution, hysteresis curve, energy consumption, and failure phenomenon of each specimen [4]. To study the low-cycle fatigue fracture performance of high-strength steel joints under strong earthquakes, Shi Yongjiong et al. completed the pseudo-static loading test of four full-scale test specimens of bolt-welded hybrid beam–column joints of high-strength steel frames, the bearing performance, and failure mode of each specimen [5]. Huang Hai et al. designed a scale model with a similar ratio of 1:4 for a complex steel joint, conducted a monotonic static loading test, studied the test phenomenon, measured the strain and failure mode, and obtained weak parts of the joint [6]. Based on these results, they developed a model test design method for complex nodes. Xia Yongqiang et al. took the T-stub as the research object, assumed the flange of the T-stub as a simply supported beam, established the initial rotational stiffness calculation model of the T-stub in the elastic stage, derived the calculation formula of the initial rotational stiffness, and verified the formula using numerical simulation and experimental data [7].

It can be seen that in the research process of many scholars, the impact of similar errors on the test results is rarely separately considered. The known research on similarities is insufficient and isolated [8–10]. Therefore, to study the mechanism of similar error formation of semi-rigid nodes, this paper derives the conditions of complete similarity of semi-rigid nodes based on similarity theory. From the perspective of end-plate thickness, the influence of similar factors was analyzed, and the corresponding influence laws were obtained. Accurate finite element models were established, and a linear regression analysis was performed to predict similar errors of semi-rigid connections.

## 2. Methods

### 2.1. Derivation of Completely Similar Macro Conditions for Semi-Rigid Connections

This chapter analyzes the unique characteristics of semi-rigid connections and derives the completely similar conditions for semi-rigid steel nodes [11]. This is the theoretical basis for further similar error analysis.

The 'so-called' macro similarity means that the target physical quantity in the derivation process is a macro quantity. For example, the dimension parameter L refers to all geometric properties and physical quantities of the node, including length, width, plate thickness, and aperture. At the same time, the stress includes the stress in a series of components, such as steel beams, steel columns, and joint core areas.

First, the main physical quantities involved in the physical process according to the analysis elements are listed in Table 1. All relevant quantities in the analysis should be included. During the static test of semi-rigid beam–column joints, the physical quantities involved are summarized as follows:

**Table 1.** Table of the physical quantities.

| Physical Quantity 1 | Physical Quantity 2 |
|---|---|
| $L$: geometric dimension. | $A$: sectional area. |
| $g$: acceleration of gravity. | $\gamma$: Poisson ratio. |
| $F$: static load. | $\rho$: density. |
| $M$: moment | $\sigma$: stress. |
| $E$: elastic modulus. | $\varepsilon$: strain. |
| $G$: shear modulus. | $s$: displacement. |
| $K$: initial stiffness of semi-rigid joint. | $\theta$: angle. |
| $I$: moment of inertia | $T$: time. |

Sixteen physical quantities were selected (shown above). This research chose F L T, which are commonly used in mechanical systems, as the basic dimensional system. The a symbol introduced by Maxwell has been used to represent the dimension of any quantity 'a'. Table 2 shows all the relevant system parameters used in the case study.

**Table 2.** Basic dimensions of related parameters.

| Basic Dimension | L | g | F | M | E | G | K | I |
|---|---|---|---|---|---|---|---|---|
| F | 0 | 0 | 1 | 1 | 1 | 1 | 1 | 1 |
| L | 1 | 1 | 0 | 1 | −2 | −2 | 1 | 2 |
| T | 0 | −2 | 0 | 0 | 0 | 0 | 0 | 1 |
| **Basic Dimension** | **A** | **γ** | **ρ** | **σ** | **ε** | **s** | **θ** | **T** |
| F | 0 | 0 | 1 | 1 | 0 | 0 | 0 | 0 |
| L | 2 | 0 | −4 | −2 | 0 | 1 | 0 | 0 |
| T | 0 | 0 | 2 | 0 | 0 | 0 | 0 | 1 |

According to the second law of similitude theory (Buckingham $\pi$ theorem), if there is a physically meaningful equation involving a certain number ($n$) of physical variables, then the original equation can be rewritten in terms of a set of $p = n - k$ dimensionless parameters constructed from the original variables.

$$f(\pi_1, \pi_2, \cdots, \pi_{n-k}) = 0 \tag{1}$$

The number of basic physical quantities is 3. Then $n = 16$, $k = 3$, and $n - k = 13$, and the above formula can be rewritten as:

$$f(M, E, G, K, I, A, \gamma, \rho, \sigma, \varepsilon, s, \theta, T) = 0 \tag{2}$$

According to the principle of dimensional homogeneity, the dimensions of both sides of the equations should be the same. Assuming the power exponent as $x_1$, $x_2$, $x_3$, and $x_4$, the equation representing $M$ (moment) in the form of basic parameters is expressed as followed:

$$L^{x_1} \cdot (LT^{-2})^{x_2} \cdot F^{x_3} = (F \cdot L)^{x_4} \tag{3}$$

Since the equation is harmonious around the dimension, sorting out the relationship gives:

$$L^{x_1 + x_2} \cdot T^{-2x_2} \cdot F^{x_3} = F^{x_4} \cdot L^{x_4} \tag{4}$$

Make the power exponent of both sides equal,

$$\begin{cases} x_3 = x_4 \\ x_1 + x_2 = x_4 \\ -2 \cdot x_2 = 0 \end{cases} \tag{5}$$

Solve the equations, assuming $x_1 = 1$, thus,

$$x_2 = 0 \tag{6}$$

$$x_1 = x_3 = x_4 \tag{7}$$

$$L \cdot F = M \tag{8}$$

From this, the first dimensionless number can be calculated:

$$\pi_1 = \frac{M}{L \cdot F} \tag{9}$$

The equations of the remaining 12 dimensionless numbers are:

$$
\begin{cases}
L^{x_{11}} \cdot (L \cdot T^{-2})^{x_{12}} \cdot F^{x_{13}} = (F \cdot L^{-2})^{x_{14}} \\
L^{x_{21}} \cdot (L \cdot T^{-2})^{x_{22}} \cdot F^{x_{23}} = (F \cdot L^{-2})^{x_{24}} \\
L^{x_{31}} \cdot (L \cdot T^{-2})^{x_{32}} \cdot F^{x_{33}} = (F \cdot L)^{x_{34}} \\
L^{x_{41}} \cdot (L \cdot T^{-2})^{x_{42}} \cdot F^{x_{43}} = (F \cdot L^2 \cdot T)^{x_{44}} \\
L^{x_{51}} \cdot (L \cdot T^{-2})^{x_{52}} \cdot F^{x_{53}} = (L^2)^{x_{54}} \\
L^{x_{61}} \cdot (L \cdot T^{-2})^{x_{62}} \cdot F^{x_{63}} = 1 \\
L^{x_{71}} \cdot (L \cdot T^{-2})^{x_{72}} \cdot F^{x_{73}} = (F \cdot L^{-4} \cdot T^2)^{x_{74}} \\
L^{x_{81}} \cdot (L \cdot T^{-2})^{x_{82}} \cdot F^{x_{83}} = (F \cdot L^{-2})^{x_{84}} \\
L^{x_{91}} \cdot (L \cdot T^{-2})^{x_{92}} \cdot F^{x_{93}} = 1 \\
L^{x_{101}} \cdot (L \cdot T^{-2})^{x_{102}} \cdot F^{x_{103}} = (L)^{x_{104}} \\
L^{x_{111}} \cdot (L \cdot T^{-2})^{x_{112}} \cdot F^{x_{113}} = 1 \\
L^{x_{121}} \cdot (L \cdot T^{-2})^{x_{122}} \cdot F^{x_{123}} = (T)^{x_{124}}
\end{cases}
\tag{10}
$$

The 13 dimensionless numbers of beam–column joints in semi-rigid steel structures can be calculated with:

$$
\begin{cases}
\pi_1 = \frac{M}{L \cdot F} \\
\pi_2 = \frac{L^2 \cdot E}{F} \\
\pi_3 = \frac{L^2 \cdot G}{F} \\
\pi_4 = \frac{K}{L \cdot F} \\
\pi_5 = \frac{g \cdot I^2}{L^5 \cdot F^2} \\
\pi_6 = \frac{A}{L^2} \\
\pi_7 = \gamma \\
\pi_8 = \frac{F}{L^3 \cdot g \cdot \rho} \\
\pi_9 = \frac{L^2 \cdot \sigma}{F} \\
\pi_{10} = \varepsilon \\
\pi_{11} = \frac{s}{L} \\
\pi_{12} = \theta \\
\pi_{13} = \frac{g \cdot t^2}{L}
\end{cases}
\Rightarrow
\begin{cases}
S_M = \frac{M_m}{M_p} \\
S_E = \frac{E_m}{E_p} \\
S_G = \frac{G_m}{G_p} \\
S_K = \frac{K_m}{K_p} \\
S_I = \frac{I_m}{I_p} \\
S_A = \frac{A_m}{A_p} \\
S_\gamma = \frac{\gamma_m}{\gamma_p} \\
S_\rho = \frac{\rho_m}{\rho_p} \\
S_\sigma = \frac{\sigma_m}{\sigma_p} \\
S_\varepsilon = \frac{\varepsilon_m}{\varepsilon_p} \\
S_s = \frac{s_m}{s_p} \\
S_\theta = \frac{\theta_m}{\theta_p} \\
S_t = \frac{t_m}{t_p}
\end{cases}
\Rightarrow
\begin{cases}
\frac{S_M}{S_L \cdot S_F} = 1 \\
\frac{S_L^2 \cdot S_E}{S_F} = 1 \\
\frac{S_L^2 \cdot S_G}{S_F} = 1 \\
\frac{S_K}{S_L \cdot S_F} = 1 \\
\frac{S_g \cdot S_I^2}{S_L^5 \cdot S_F^2} = 1 \\
\frac{S_A}{S_L^2} = 1 \\
S_\gamma = 1 \\
\frac{S_F}{S_L^3 \cdot S_g \cdot S_\rho} = 1 \\
\frac{S_L^2 \cdot S_\sigma}{S_F} = 1 \\
S_\varepsilon = 1 \\
\frac{S_x}{S_L} = 1 \\
S_\theta = 1 \\
\frac{S_g \cdot S_t^2}{S_L} = 1
\end{cases}
\tag{11}
$$

The above equation shows the exact similar conditions of the beam–column joint of the semi-rigid steel structure. Assuming that the geometric scale ratio of the model is 1:2, according to Equation (11), the similarity ratio of each physical quantity in Table 3 can be obtained. In a typical semi-rigid steel structure node test, the controlled test conditions are numerous. First, the similarity ratios of size parameters cannot all be consistent. When the size of a local component cannot meet the dimensional similarity ratio, the similarity error caused cannot be derived from traditional similarity conditions. This article focuses on the effect of inconsistent end-plate thickness on similar errors.

**Table 3.** Similarity ratio of each factor.

| Factor | Similarity Ratio | Factor | Similarity Ratio |
|--------|------------------|--------|------------------|
| $S_L$  | 1:2              | $S_A$  | 1:4              |
| $S_F$  | 1:4              | $S_\gamma$ | 1:1          |
| $S_M$  | 1:8              | $S_\sigma$ | 1:1          |
| $S_E$  | 1:1              | $S_\varepsilon$ | 1:1     |
| $S_G$  | 1:1              | $S_s$  | 1:2              |
| $S_K$  | 1:8              | $S_\theta$ | 1:1          |

### 2.2. Numerical Simulation Method

Many deficiencies in the similarity relationships derived from the classic similarity theory have not been able to fully meet the development requirements of current model tests [12–20]. To solve this problem, ABAQUS 6.14 (2014, Dassault SIMULIA, Inc., Providence, USA) was used to establish an accurate node finite element model to simulate similar errors. The element type of the finite element model in this paper is the linear reduced integral element C3D8R, the grid control size of the beam–column member is 15 mm, the end-plate is 4 mm, and the bolt is 1.4 mm. The model in this paper adopts the loading method of displacement control. The analysis in this article is a static analysis, so the constitutive relationship of steel uses a trilinear model. The strength grade of the beam and column steel is Q345B, the ultimate strain is 0.1, and the elastic modulus is taken as 206,000 MPa. The bolts are all 10.9 high-strength friction-type bolts, the ultimate strain is 0.1, and the elastic modulus is 210,000 MPa. The size parameters of the model are shown in Table 4 and Figure 1.

**Table 4.** Component modulus of completely similar conditions.

| Prototype | Component Modulus (mm) | Model | Component Modulus of Completely Similar Conditions (mm) |
|---|---|---|---|
| $B_p$ | HN300 × 200 × 8 × 12 | $B_m$ | 150 × 100 × 4 × 6 |
| $C_p$ | HN300 × 300 × 10 × 15 | $C_m$ | 150 × 150 × 5 × 7.5 |
| $E_p$ | 250 × 500 × 16 | $E_m$ | 125 × 250 × 8 |
| $BL_p$ | M20 | $BL_m$ | M10 |

$B_p$: beam of prototype; $C_p$: column of prototype; $E_p$: end plate of prototype; $BL_p$: bolt of prototype; $B_m$: beam of model; $C_m$: column of model; $E_m$: end-plate of model; $BL_m$: bolt of model. HN: Narrow flange H-beam. M: Bolt specifications.

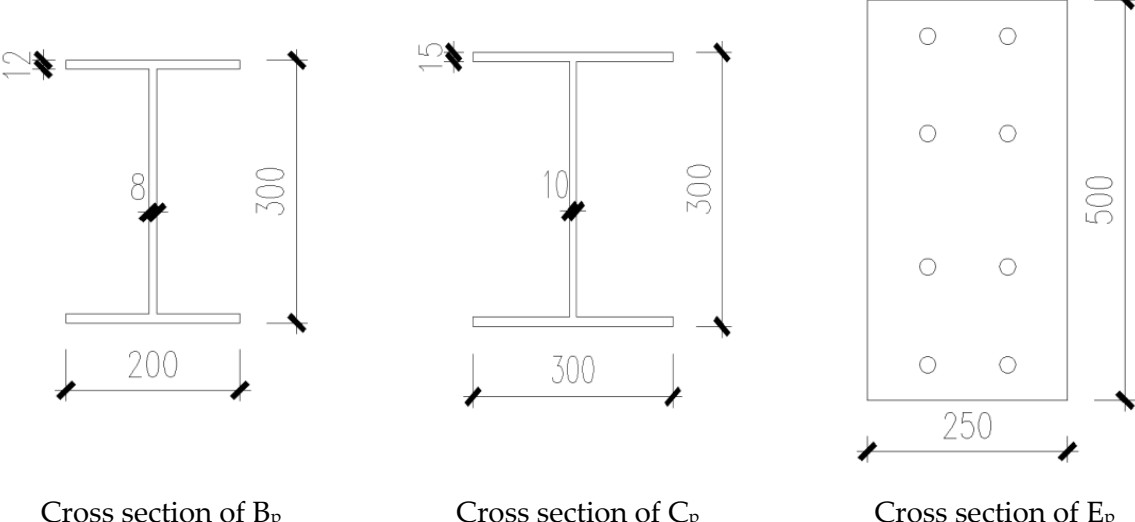

|  |  |  |
|---|---|---|
| Cross section of $B_p$ | Cross section of $C_p$ | Cross section of $E_p$ |

**Figure 1.** Schematic diagram of component dimensions.

To make the simulation as close as possible to the actual situation, fine modeling was performed at the boundary of the nodes in the finite element analysis simulation, including ribs and stiffeners, which are consistent with the actual conditions of the actual test. Column length is the distance between the inflection points of the upper and lower adjacent layers in the actual structure. The model adopts the form of column-head loading and beam-end restraint, the bottom of the column is simulated as a hinged bearing constraint, and the beam ends are telescopic sliding bearings. The contact surface of the column and the end-plate was set to contact. The definition of the contact properties was divided into two parts. The normal action was defined as the hard contact type, the tangential action was defined as the Coulomb friction type, and the friction coefficient was defined as 0.4. The model is shown in Figure 2. The boundary conditions are shown in Figure 3. Two materials were used in the

finite element model, which were high-strength bolts and beam–column steel. The material properties of the two steels are shown in Table 5.

In order to avoid the simulation error caused by the size effect of the unit, the unit size ratio of the model and the prototype is set to a ratio that meets the similar ratio, see Table 6 for details. By comparing the stress distribution of the completely similar model and prototype, the accuracy of numerical simulation is illustrated.

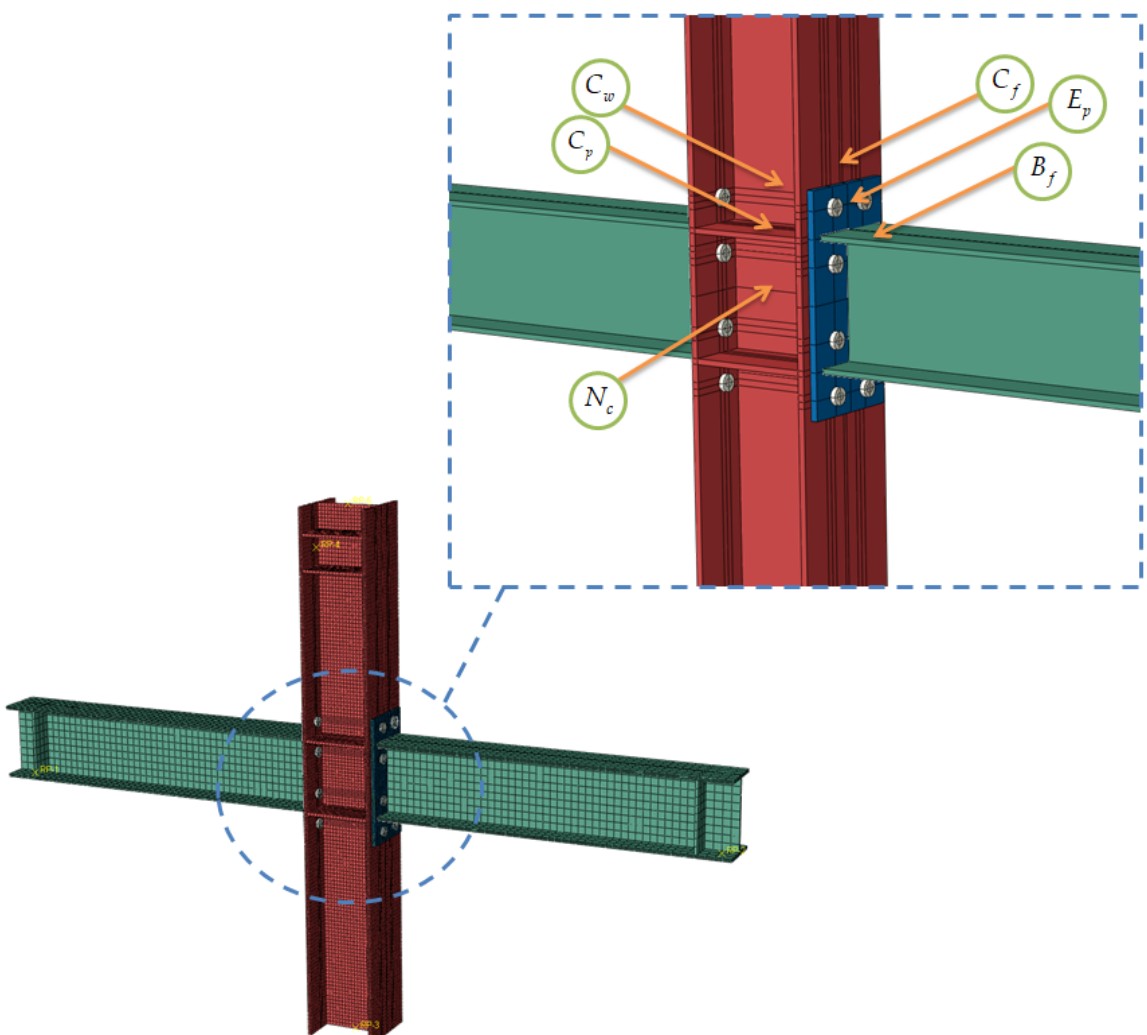

**Figure 2.** Specified points of the finite element model.

Note: $B_f$: the upper flange of the beam and 75 mm away from the end-plate; $C_f$: the flange of the column and 75 mm away from the top of the beam; $C_p$: the central position of the upper partition of the column; $C_w$: the web of the column and 75 mm away from the surface of the upper partition; $N_c$: the central position of the core area of the node; and $E_p$: the upper and middle parts of the end-plate 27.5 mm away from the top of the beam.

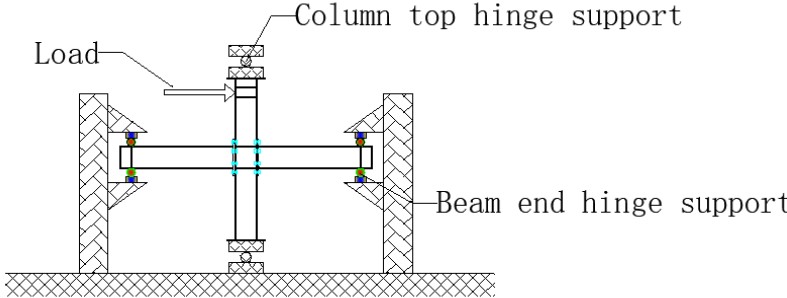

**Figure 3.** Boundary conditions of the model.

**Table 5.** Material properties of two steels.

| Index | Bolt | Beam and Column |
|---|---|---|
| Yield strength | 940 | 373.2 |
| Ultimate strength | 1130 | 540.7 |
| Yield strain | 0.05 | 0.08375 |
| Ultimate strain | 0.1 | 0.3 |

**Table 6.** Model and prototype unit dimensions.

| Position | Prototype (mm) | Model (mm) |
|---|---|---|
| Beam | 30 | 15 |
| Column | 30 | 15 |
| End-plate | 8 | 4 |
| Bolt | 2.8 | 1.4 |

The dimensions in the table are the mesh sizes in the finite element models.

## 3. Results

### 3.1. Comparison of Semi-Rigid Node Indicators Under Completely Similar Conditions

This section looks for the distribution of similar errors by analyzing the errors between geometrically incompletely similar models and completely similar models. To study the error, the data of the completely similar model must be obtained first. To examine the completely similar results, a model that strictly adheres to the 1:2 geometric scale of the prototype node is established, and the parameters of the scaled model are determined based on the fully similar conditions derived. Similar conditions can be judged from the stress state of the key parts of the node.

As can be seen from Figures 4–8, the stress distributions of the nodes are almost exactly the same under the completely similar situation, including the stress distribution of beam–column members, end-plates, and the core area of the nodes. The stress data of the six typical points, such as the upper flange of the beam, the outer flange of the column, the partition on the column, the upper part of the column web, the core area of the node, and the upper and middle part of the end-plate, are listed in Figures 6–8. It can be seen that with the progress of the loading process, the stress changes of the upper flange of the beam, the core area of the joint, and the upper and middle parts of the end-plate increase monotonically, while the change trend of the stress at the outer flange of the column first decreases and then increases, and the trend of the stress change at the upper part of the column web is to maintain flatness and then increase monotonically. After superimposing the above-mentioned point stress trends of the prototype and the completely similar model (Figure 8), it can be seen that the curves of the two are almost identical, which proves the correctness of the complete similarity of the two and the previously derived completely similar conditions.

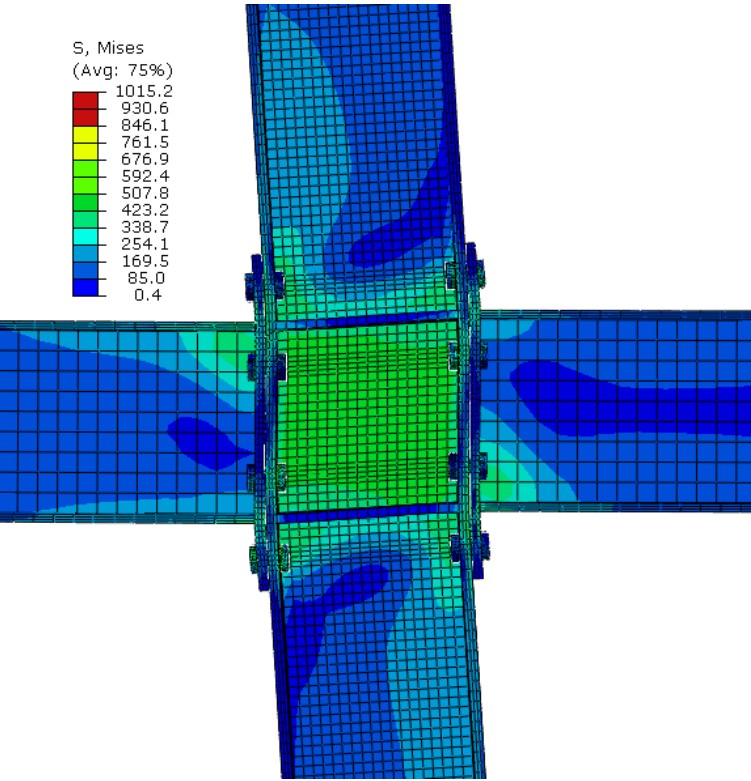

**Figure 4.** Prototype stress distribution.

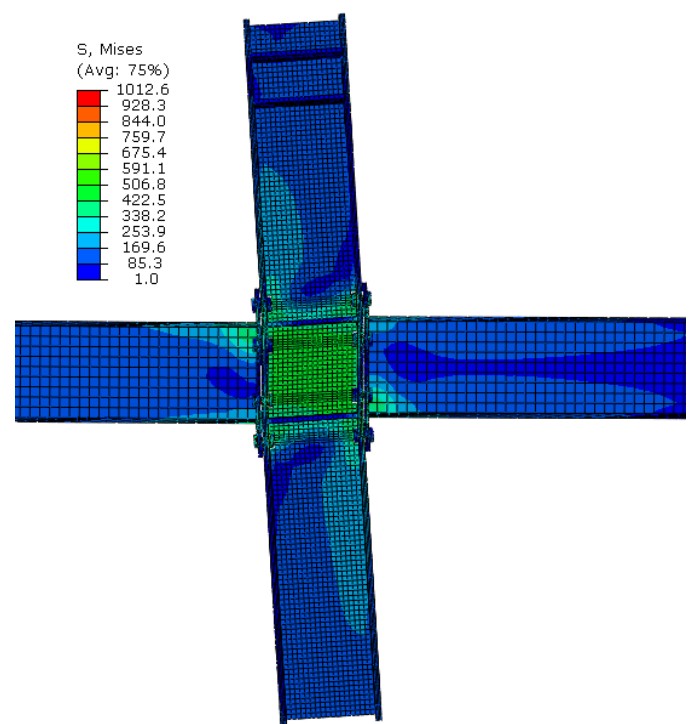

**Figure 5.** Model stress distribution.

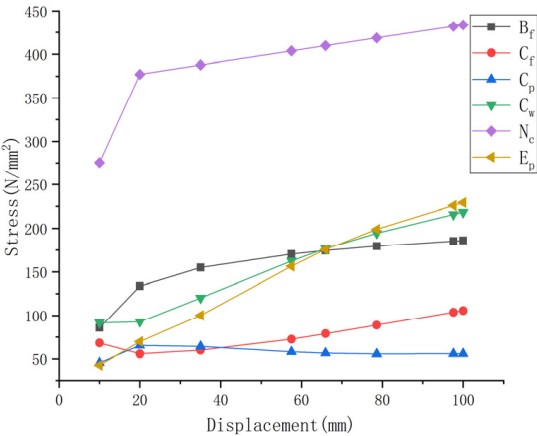

**Figure 6.** Prototype stress history.

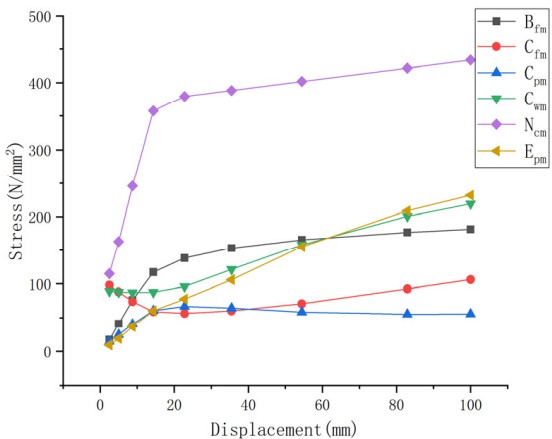

**Figure 7.** Model stress history.

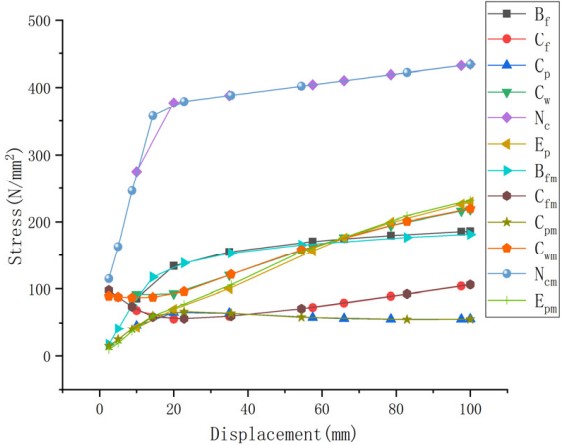

**Figure 8.** Comparison of the stress history of the prototype and the model.

### 3.2. Effect of End-Plate Thickness on Similar Errors

To study the effects of end-plate thickness on similar errors, six non-exact similar models with a 1:2 scale ratio were established under exactly the same conditions, with end-plate thicknesses of 4 mm, 6 mm, 10 mm, 12 mm, 14 mm, and 16 mm, with other similar conditions unchanged, where the joint stress distributions under different end-plate thickness conditions are listed as shown in Figures 9–21.

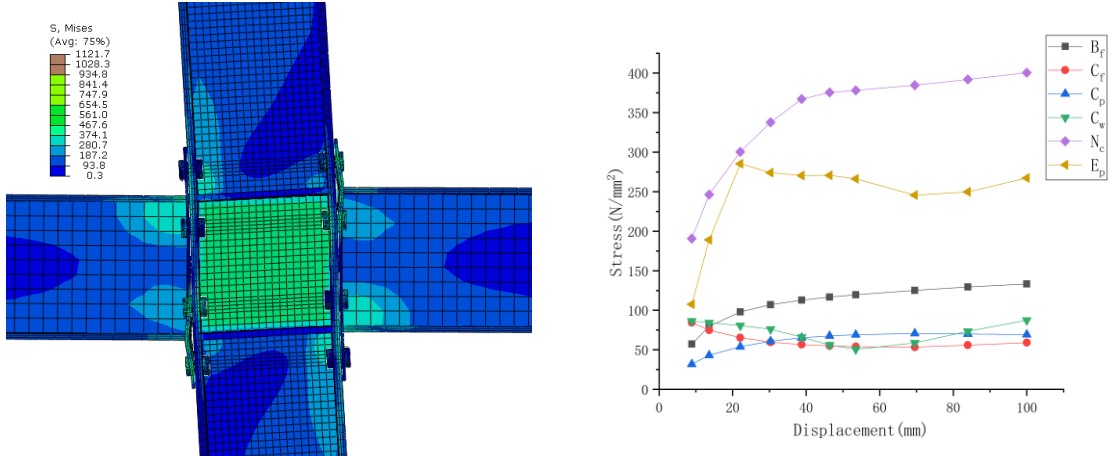

**Figure 9.** Stress distribution and stress history at specified points of a 4-mm end-plate.

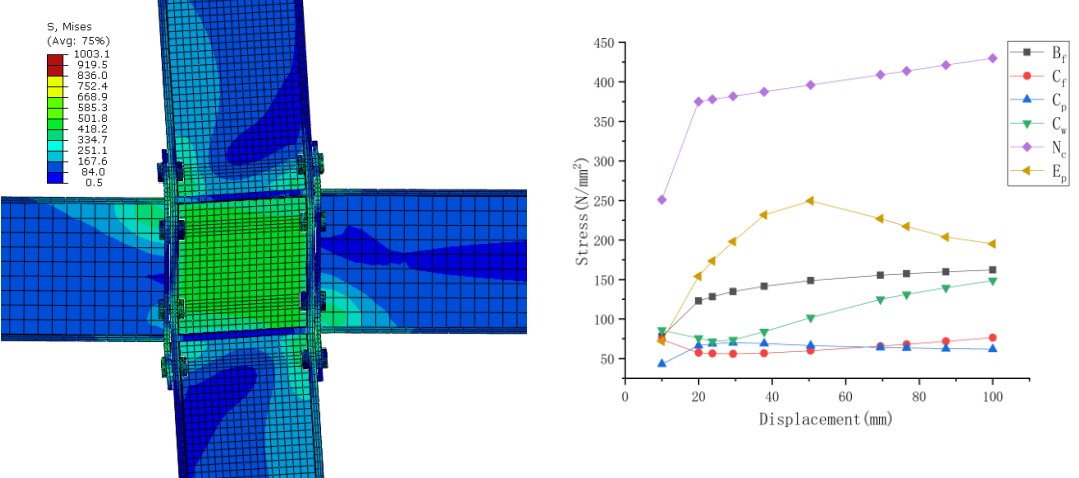

**Figure 10.** Stress distribution and stress history at specified points of a 6-mm end-plate.

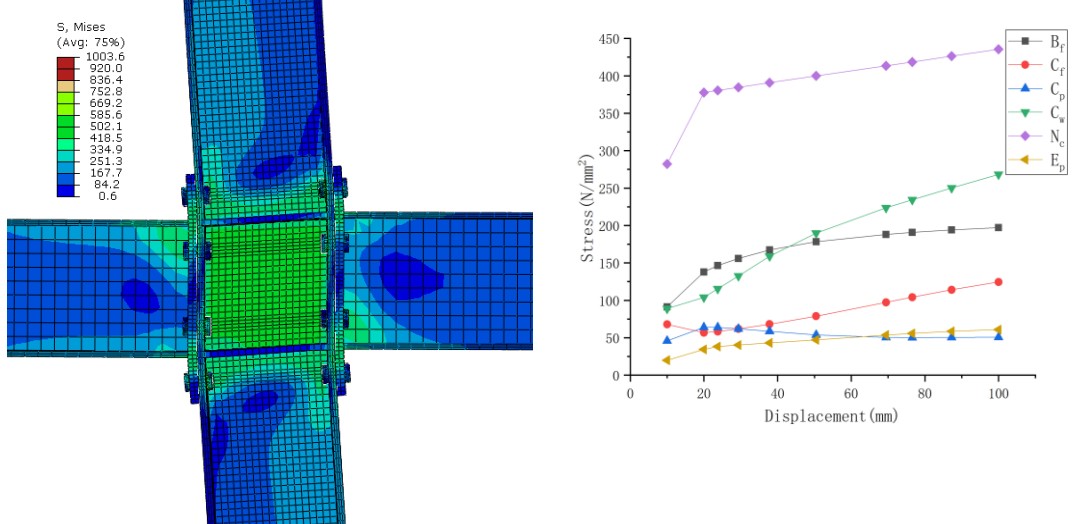

**Figure 11.** Stress distribution and stress history at specified points of a 10-mm end-plate.

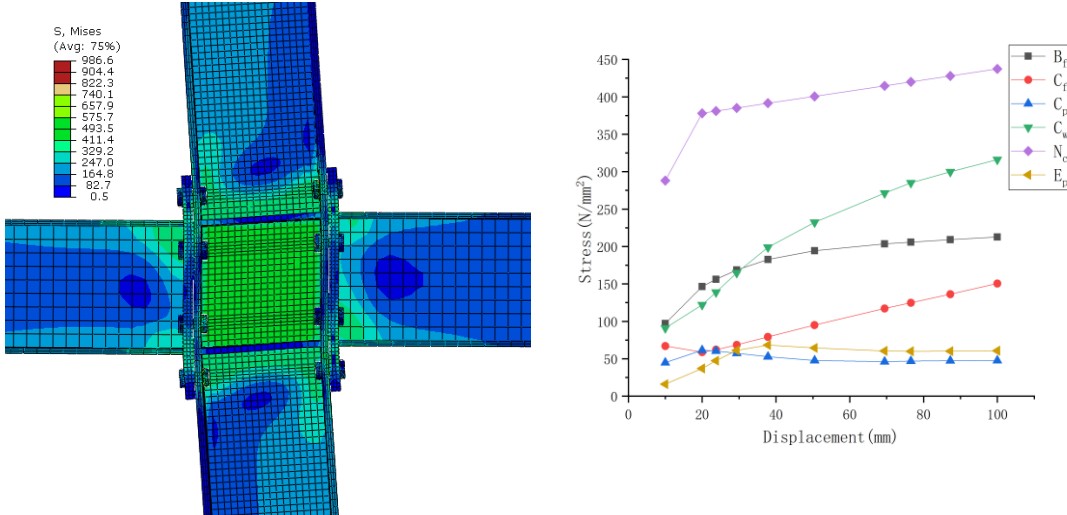

**Figure 12.** Stress distribution and stress history at specified points of a 12-mm end-plate.

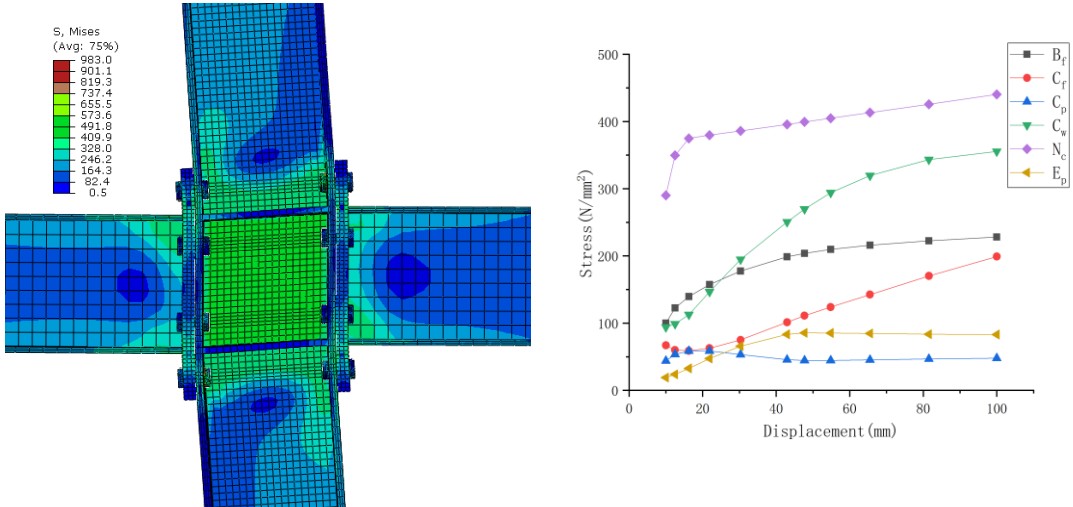

**Figure 13.** Stress distribution and stress history at specified points of a 14-mm end-plate.

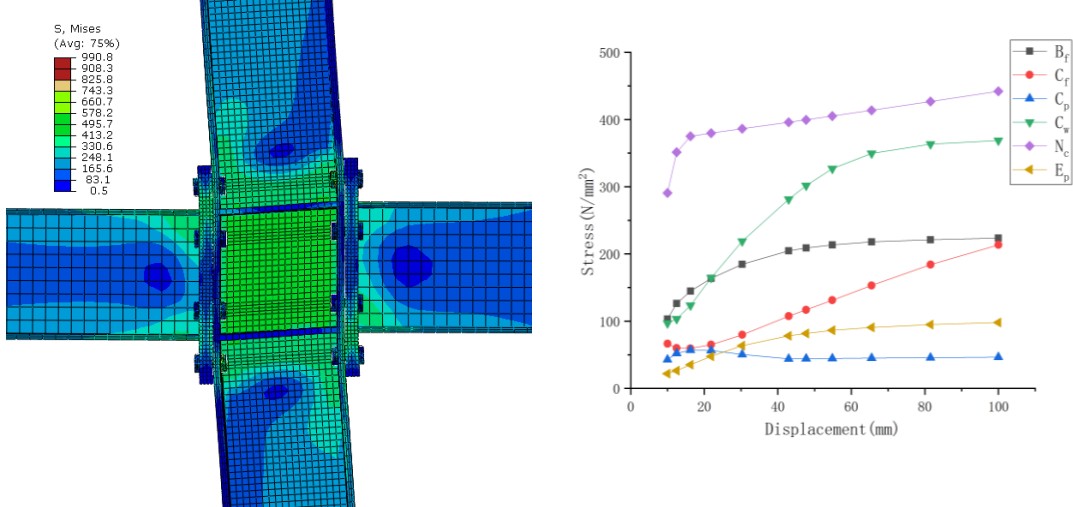

**Figure 14.** Stress distribution and stress history at specified points of a 16-mm end-plate.

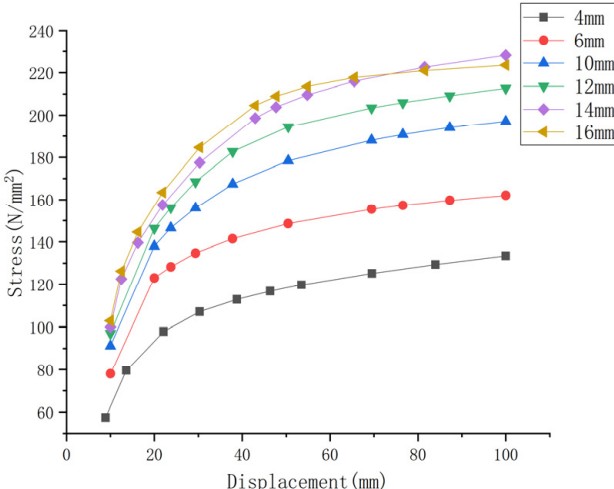

**Figure 15.** Stress history at $B_f$.

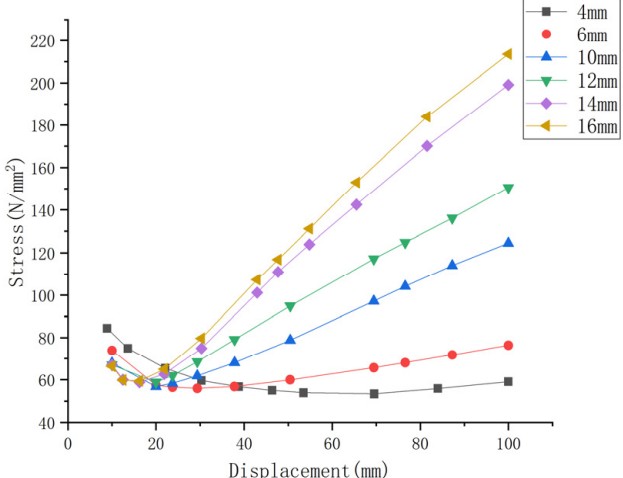

**Figure 16.** Stress history at $C_f$.

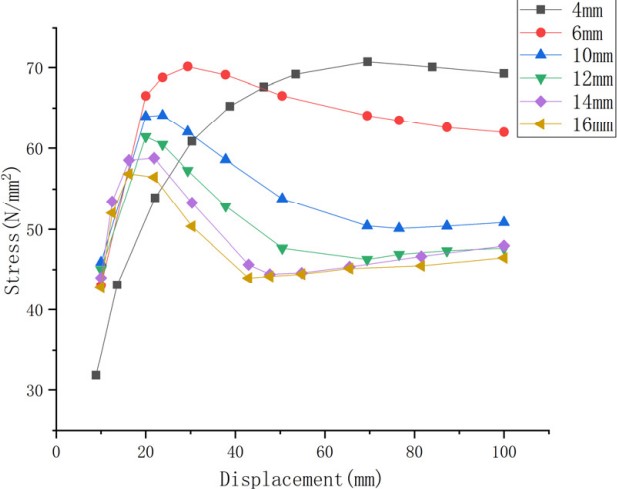

**Figure 17.** Stress history at $C_p$.

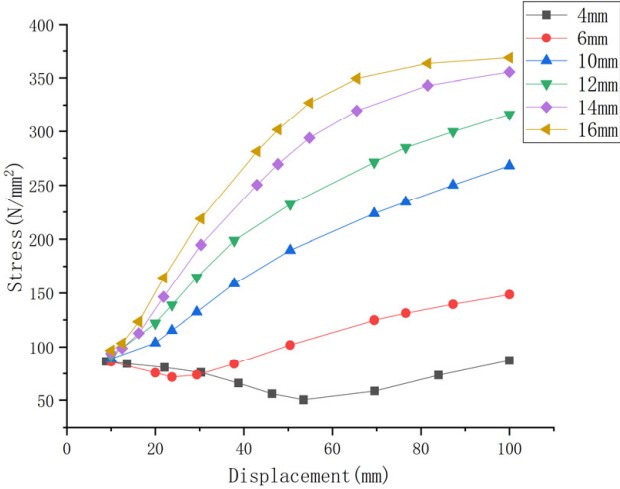

**Figure 18.** Stress history at $C_w$.

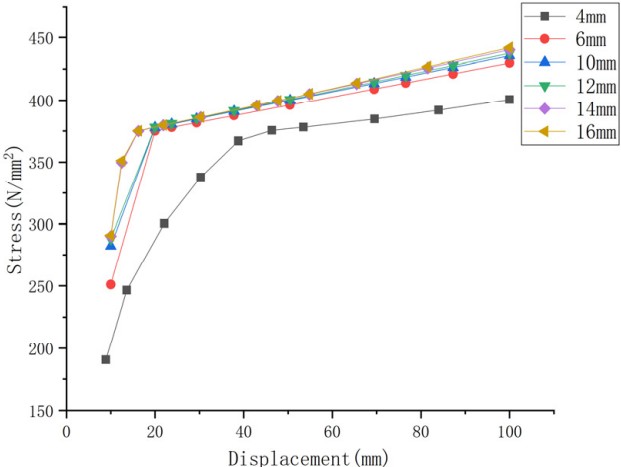

**Figure 19.** Stress history at $N_c$.

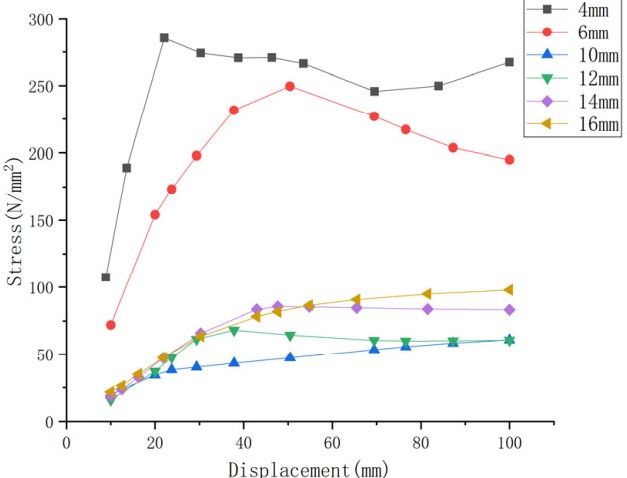

**Figure 20.** Stress history at $E_p$.

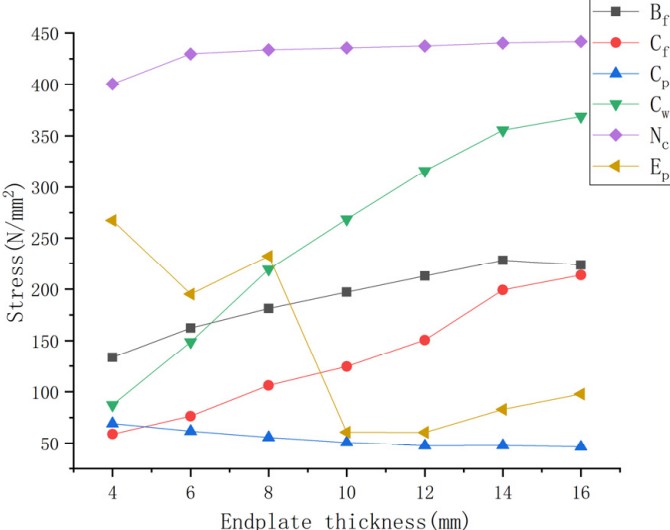

**Figure 21.** Comparison of stress history at specified points.

As can be seen in Figures 9–21, compared with the stress distribution of completely similar models, when the end-plate thickness deviates from the completely similar conditions, the model's stress distribution shows a large difference. The stress trends and numerical levels of beam and column members, end-plate, and the core of the joint are no longer highly consistent with the prototype, and a large similarity error has occurred. The six typical points on the beam flange, including the outer flange of the column, the partition on the column, the upper part of the column web, the core area of the node, and the upper and middle parts of the end-plate, are still selected. The changes in stress data are listed in Figures 9–21. It can be seen that if the degree of deviation from the completely similar model is different, the value and change law of the stress distribution at each point will be considerably different.

For the upper flange of the beam, the trend of stress changes is monotonically increasing when the thickness of the end-plate is 16 mm and the stress decreases in the later stage of loading, which is inconsistent with the overall law. For the outer flange of the column, the trend of the stress change is to decrease first and then increase, and when the thickness of the end-plate is 4 mm, the stress growth in the later stage is relatively slow and all other factors linearly increase. For the partition on the column, the trend of stress changes is first increasing, then decreasing, and then flattening, but when the end-plate thickness is 4 mm, the stress does not decrease. For the upper part of the column web, except for the end-plate thicknesses of 4 mm and 6 mm, the stress change trend is monotonous, and the deviation from the overall trend is more obvious at 4 mm. For the core area of the node, the trend of stress change is a monotonic increase, except for the 4 mm case, for which the changes of the other curves are closer. For the middle and upper parts of the end-plate, when the thickness is 4 mm and 6 mm, the stress change trend is to increase first and then decrease, and the rest are monotonous.

Therefore, when the thickness of the end-plate deviates far from the complete similarity, the value of stress and the law of change will change greatly, especially when the thickness of the end-plate is too thin or too thick; the rule will be more obvious; and the stress change trend of the loading process will significantly change.

Through the stress change diagram of the end-plate connection with the thickness of the end-plate (Figure 22), it can be seen that as the thickness of the end-plate increases, the stress distribution of the end-plate changes significantly and presents a nonlinear characteristic. In the case of 4mm thickness, the maximum stress is mainly concentrated around the bolt hole. In the case of 16mm thickness, the maximum stress distribution range is obviously expanded, and the maximum stress gradually decreases as the thickness of the end-plate increases. Therefore, although the stress value of the end-plate does not change linearly with the increase in thickness, it is regular. In this paper, the linear regression method is not used to accurately simulate the distribution law of node stress, but to reduce

the similar error according to the general data law. Through linear regression, it can be seen that the similarity error of the end-plate is greatly reduced. Linear regression is not a perfect method for the error analysis involved in this article, but it is an effective method.

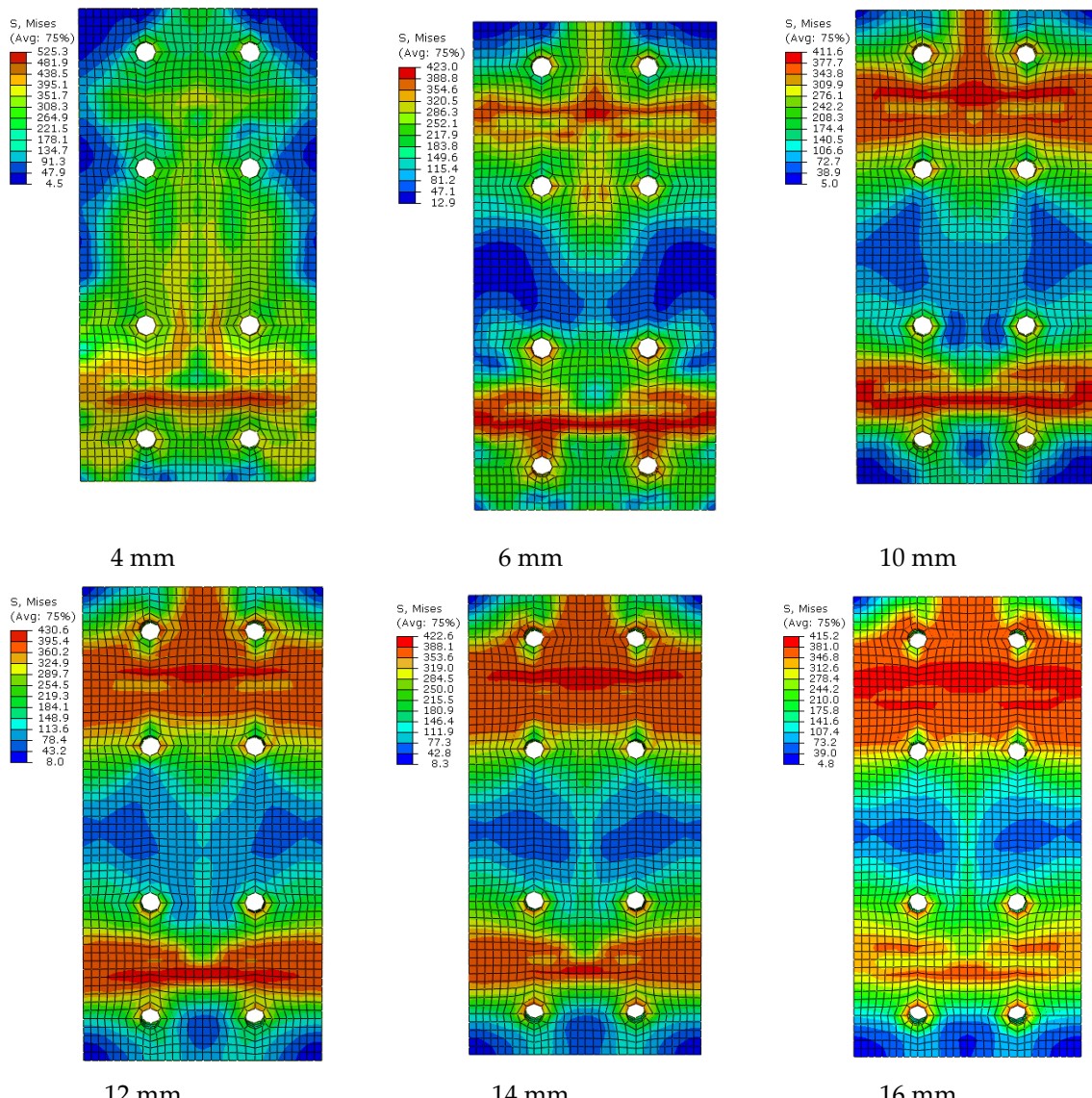

**Figure 22.** Stress distribution of the end-plate.

## 4. Error Analysis Method

Through parameterized numerical model analysis, we obtained a large number of stress data of incompletely similar nodes. Based on these data, this paper used regression analysis to obtain similar errors. The flow chart of modeling and regression analysis of incompletely similar nodes is as follows (Figure 23):

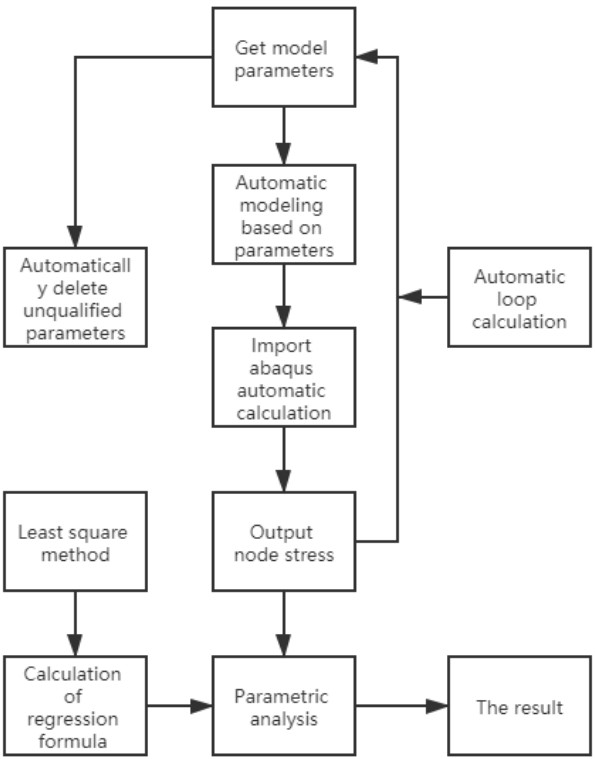

**Figure 23.** Analysis flow chart.

The method of regression analysis is as follows:

Regression analysis is a predictive modeling technique, which studies the relationship between a dependent variable (target) and an independent variable (feature). This technique is often used for predictive analysis and to find causal relationships between variables. Usually a curve is used to fit the data points, and the goal is to minimize the difference in distance between the curve and the data points. Linear regression is a regression problem which assumes that the target value and the characteristics are linearly related, that is, to satisfy a multivariate linear equation. By constructing the loss function, the parameters w and b when the loss function is minimum are solved. Usually, we can express it as the following formula:

$$y = wx + b \tag{12}$$

where $y$ is the predicted value, and the independent variable $x$ and the dependent variable $y$ are known. What we want to achieve is the prediction of the value of $y$ when adding a new $x$. Therefore, in order to construct this functional relationship, we need to solve the two parameters $w$ and $b$ in the linear model through known data points.

Solving the best parameters requires a standard to measure the results. For this, we need to quantify an objective function formula so that the computer can continuously optimize during the solution process.

For the error-solving problem of incompletely similar nodes, the loss function can be defined as follows:

$$L = \frac{1}{n} \sum_{i=1}^{n} \left(y_{pi} - y_i\right)^2 \tag{13}$$

where $y_{pi}$ is the predicted value of the node, $y_i$ is the known value of the node, and $n$ is the number of node models.

$L$ is the average squared distance between the predicted value and the true value. Substituting Equation (11) into the loss function, and taking the parameters $w$ and $b$ to be solved as the independent variables of the function $L$, we can obtain:

$$L(w, b) = \frac{1}{n} \sum_{i=1}^{n} (wx_i + b - y_i)^2 \tag{14}$$

In order to obtain the optimal solutions of $w$ and $b$, it is necessary to minimize the loss function $L$. Using the least squares parameter estimation method, we can derive $L(w,b)$ to $w$ and $b$, respectively. Suppose the derivative is 0 and calculate the closed solution of the optimal solution of $w$ and $b$:

$$w = \frac{\sum_{i=1}^{n} y_i \left( x_i - \bar{x} \right)}{\sum_{i=1}^{n} x_i{}^2 - \frac{1}{n} \left( \sum_{i=1}^{n} x_i \right)^2} \tag{15}$$

$$b = \frac{1}{n} \sum_{i=1}^{n} (y_i - wx_i) \tag{16}$$

## 5. Discussion

The above analysis shows that when the thickness of the end-plate changes, the overall stress error of incompletely similar nodes cannot be ignored, and there is a certain regularity. A regression analysis was performed on the error data to obtain a similar error calculation method. The error data were arranged based on the calculations in Figure 21.

It can be seen in Figure 24 that all the error data, except for the middle and upper parts of the end-plate, present a monotonic linear growth relationship related to the end-plate thickness. Therefore, the data distribution is suitable for linear regression analysis. By importing the data into Equations (12)–(16) for univariate linear regression analysis, we obtain the corresponding regression formula and correlation coefficient, the closer the correlation coefficient was to 1, the more significant the regression formula. The comparison between the regression curve and the original data points is shown in Figures 25–30. Figure 31 lists the reduction degree of similar error after linear regression.

The regression formula of $B_f$: $Y = -0.3727 + 0.0428x$. The correlation coefficient: 0.9349.

The regression formula of $C_f$: $Y = -1.0157 + 0.1265x$. The correlation coefficient: 0.9823.

The regression formula of $C_p$: $Y = 0.3083 - 0.0335x$. The correlation coefficient: 0.8817.

The regression formula of $C_w$: $Y = -0.9547 + 0.1104x$. The correlation coefficient: 0.9699.

The regression formula of $N_c$: $Y = -0.0667 + 0.0061x$. The correlation coefficient: 0.6579.

The regression formula of $E_p$: $Y = 0.3089 - 0.0695x$. The correlation coefficient: 0.6445.

Note $B_f$: the upper flange of the beam and 75 mm away from the end-plate; $C_f$: the flange of the column and 75 mm away from the top of the beam; $C_p$: the central position of the upper partition of the column; $C_w$: the web of the column and 75 mm away from the surface of the upper partition; $N_c$: the central position of the core area of the node; and $E_p$: the upper and middle parts of the end-plate 27.5 mm away from the top of the beam.

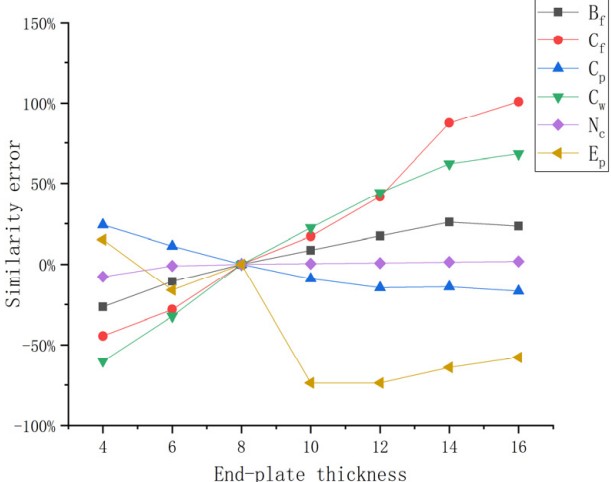

**Figure 24.** Similarity errors of different end-plate thicknesses at specified points.

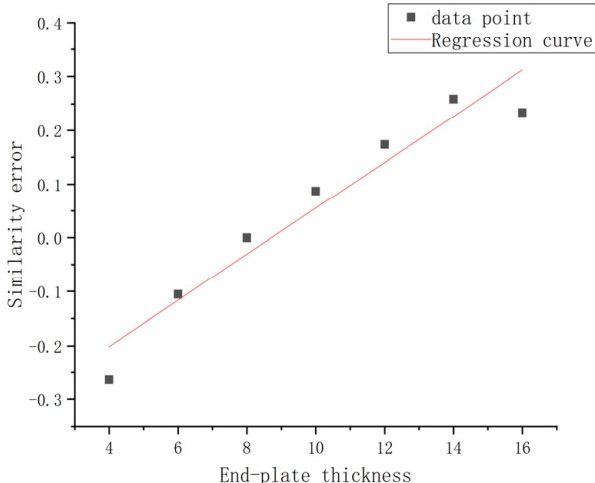

**Figure 25.** Data points and the regression curve at $B_f$.

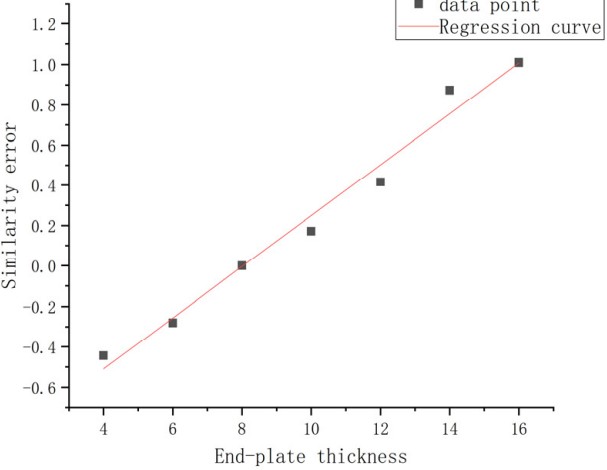

**Figure 26.** Data points and the regression curve at $C_f$.

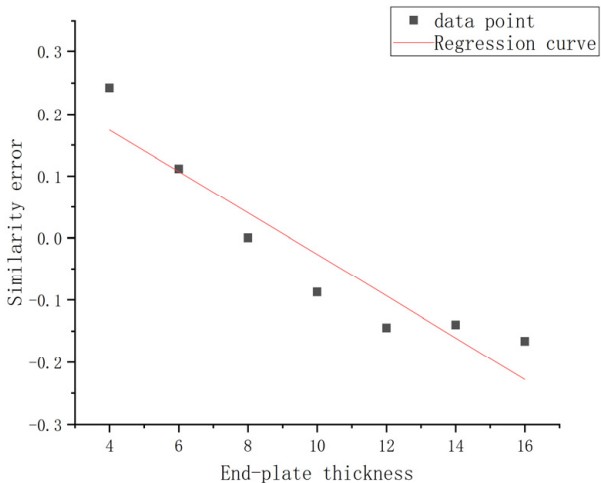

**Figure 27.** Data points and the regression curve at $C_p$.

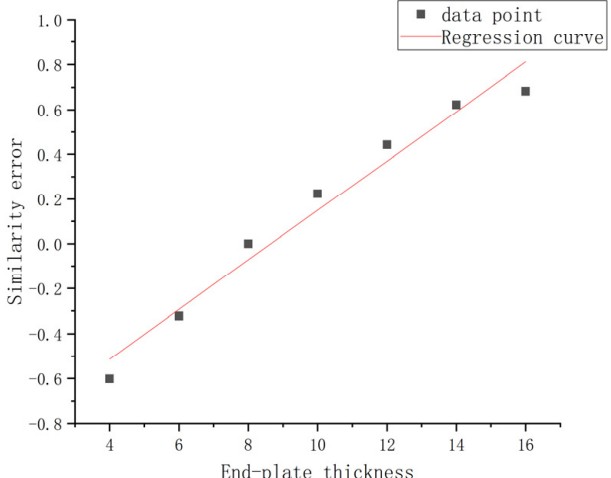

**Figure 28.** Data points and the regression curve at $C_w$.

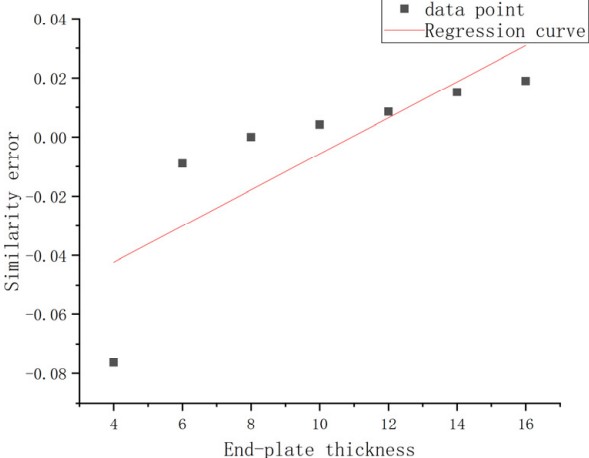

**Figure 29.** Data points and the regression curve at $N_c$.

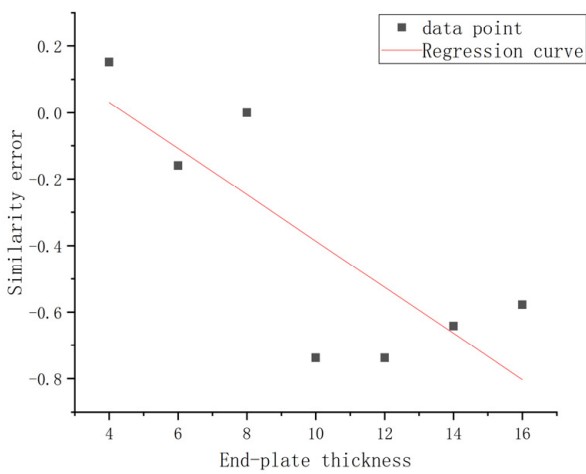

**Figure 30.** Data points and the regression curve at $E_p$.

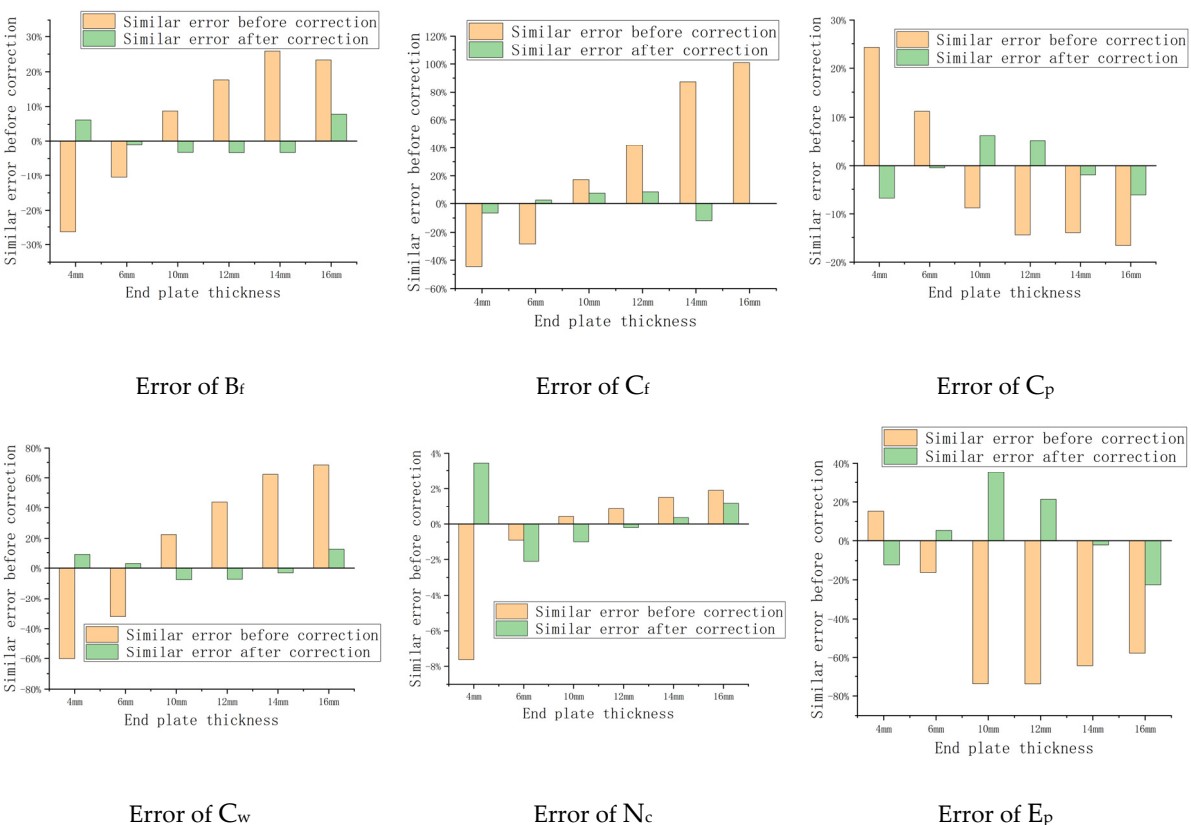

**Figure 31.** Linear regression-corrected value.

Therefore, the linear regression can realize the prediction and correction of most similar errors, and the residual similar errors after correction are shown in Table 7. Except for the error of the stress data in the middle and upper parts of the end-plate, most of the similar errors have been greatly improved. In this way, the similarity error problem of incompletely similar models has been solved.

**Table 7.** The correction of similarity error by fitting algorithm.

| Name | Status | 4 (mm) | 6 (mm) | 8 (mm) | 10 (mm) | 12 (mm) | 14 (mm) | 16 (mm) |
|---|---|---|---|---|---|---|---|---|
| $B_f$ | before | −26.38% | −10.55% | 0.00% | 8.75% | 17.40% | 25.89% | 23.35% |
| | after | 6.23% | −1.04% | −3.03% | −3.22% | −3.31% | −3.24% | 7.86% |
| $C_f$ | before | −44.40% | −28.07% | 0.00% | 17.22% | 41.65% | 87.37% | 100.94% |
| | after | −6.57% | 2.40% | −0.37% | 7.71% | 8.58% | −11.84% | −0.11% |
| $C_p$ | before | 24.24% | 11.18% | 0.00% | −8.78% | −14.46% | −14.00% | −16.62% |
| | after | −6.81% | −0.45% | 4.03% | 6.11% | 5.09% | −2.07% | −6.15% |
| $C_w$ | before | −60.19% | −32.18% | 0.00% | 22.44% | 44.27% | 62.22% | 68.29% |
| | after | 8.88% | 2.95% | −7.15% | −7.51% | −7.26% | −3.13% | 12.88% |
| $N_c$ | before | −7.64% | −0.90% | 0.00% | 0.42% | 0.85% | 1.52% | 1.90% |
| | after | 3.41% | −2.11% | −1.79% | −0.99% | −0.20% | 0.35% | 1.19% |
| $E_p$ | before | 15.17% | −15.96% | 0.00% | −73.79% | −73.85% | −64.23% | −57.81% |
| | after | −12.08% | 5.15% | −24.71% | 35.18% | 21.34% | −2.18% | −22.50% |

$B_f$: the upper flange of the beam and 75 mm away from the end-plate; $C_f$: the flange of the column and 75 mm away from the top of the beam; $C_p$: the central position of the upper partition of the column; $C_w$: the web of the column and 75 mm away from the surface of the upper partition; $N_c$: the central position of the core area of the node; and $E_p$: the upper and middle parts of the end-plate 27.5 mm away from the top of the beam.

It is the main purpose of this paper to reduce the similarity error by looking for the law of data change. The authors did not pursue accurate linear stress change laws, but attempted to reduce similar errors through linear regression. It can be seen that linear regression is at least an effective method applied in this field.

## 6. Conclusions

(1) Through the derivation, the completely similar conditions of the semi-rigid beam–column connection structure were obtained, and the constraints of the similar factors were clarified. This paper analyzed the limitations of the classic similarity theory and provided a theoretical basis for further research on similarity errors.

(2) The numerical model proves that under the premise of meeting the completely similar conditions, the prototype and the model's stress distribution law and development process are completely consistent.

(3) This paper took the thickness of the end-plate as a similar factor, established six incomplete scale models, selected the stress development history of six points as the research object, analyzed, in detail, the influence of the end-plate thickness on the model's stress development process and distribution when it does not meet the exact similar conditions, and summarized the stress development law of typical points of the end-plate connection.

(4) This paper calculated the similarity errors due to the incomplete end-plate thicknesses. The analysis shows that the similarity error and the similarity factor are roughly monotonic and linear. A regression analysis was performed, and a regression formula for similar error and end-plate thickness was obtained. The correlation coefficient of the corresponding formula was calculated, and the results show that except for the stress at the end-plate points, the univariate linear regression equations at other points are more significant. Similar error analysis methods were provided which can be applied to other types of models.

(5) The analysis results in this paper show that even if the stress distribution of the model only approximately meets the linear distribution law, the linear regression method can still reduce the incomplete similarity error to a large extent, which provides a reference for future study of the similarity error distribution law.

**Author Contributions:** Conceptualization: D.Z.; methodology, J.P.; software, D.Z.; validation, J.P. and Z.W.; formal analysis, P.W.; investigation, D.Z.; resources, J.P.; data curation, D.Z.; writing—original draft preparation, D.Z.; writing—review and editing, D.Z.; visualization, D.Z.; supervision, Z.W.; project administration, Z.W.; funding acquisition, Z.W. All authors have read and agreed to the published version of the manuscript.

**Funding:** This study was supported by the National Natural Science Foundation of China (Grant No. 51638009, 51778241, 51978279), the State Key Laboratory of Subtropical Building Science, South China University of Technology (Grant No. 2017ZB28, 2017KD22), the Fundamental Research Funds for the Central Universities, South China University of Technology (Grant No. 2019PY20, D2191390), the Chinese Postdoctoral Foundation of China (Grant No. 2019M652898), Guangdong Basic and Applied Basic Research Foundation (2020A1515011307) and the Young Innovative Talents Program in Universities and Colleges of Guangdong Province (2018KQNCX006).

**Acknowledgments:** We appreciate the linguistic assistance provided by TopEdit during the preparation of this manuscript.

**Conflicts of Interest:** The authors declare no conflict of interest.

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
