# Peer review of "Error Analysis Method of Geometrically Incomplete Similarity of End-Plate Connection Based on Linear Regression"

_applsci, doi:10.3390/app10144812_

Round 1
Reviewer 1 Report
What was the reason of using a scale 1:2 for model?
How did you established that the connection is a “semirigid- connection”?
Has the relative rotation between beam and column been established by the analysis?
From my expertise I would suggest the use of two-side flange plate (for constructive reasons).
On rows 267, 279 (in version v1) change “Table 4” in “Table 6”.
Reviewer 2 Report
In the reviewer opinion the paper entitled “ Error Analysis Method of Geometrical Incomplete Similarity of End plate Connection Based on Linear Regression” is poor. The research does not present scientific work. It rather resembles “a draft” from the data collecting stage. The main problem is that it is hard to understand what the purpose of the paper is as the objective was not clearly stated. The article has serious flaws, the research not conducted correctly and clearly. The whole investigation lacks in the general framework.
The clarification of the problems as well as the organization of the text is very poor. For example, Table 2 is even not cited in the whole text (however, this Table seems to be very important in terms of the prototype dimensions and the whole problem of similarity). Table 5 is cited before Table 4. It is not clear what studies were performed on the propotype? It is said that (lines 119-121): “The model in this paper adopts the loading method of displacement control. The analysis in this article is a static analysis, so the constitutive relationship of steel uses a trilinear model.” These sentences do not clarify the used materials & methods sufficiently. What does “Boundary conditions of the model” mean in Figure 2? The presented numerical model, created in Abaqus, lack in some important information on boundary conditions.
The only information on the methodology of the research is covered by one sentence: “By comparing the stress distribution of the completely similar model and prototype, the accuracy of numerical simulation is illustrated.” It is too less information to follow the clue.
The following text is repeated several times under Figures or Tables (the manuscript must not have so many repetitions): “Note: Bf: the upper flange of the beam and 75mm away from the end plate; Cf: the flange of the column and 75mm away from the top of the beam; Cp: the central position of the upper partition of the column; Cw: the web of the column and 75mm away from the surface of the upper partition; Nc: the central position of the core area of the node; and Ep: the upper and middle parts of the endplate and 27.5mm away from the top of the beam.”
Reviewer 3 Report
Error Analysis Method of Geometrical Incomplete Similarity of End plate Connection Based on Linear Regression
The manuscript addresses an interesting topic, with regard to the detection and minimization of errors in the study of steel beam-to-column joint models.
The manuscript is structured in 5 chapters, from the introduction to the conclusion.
As a general comment, in the abstract, it is mentioned that the article intends to propose a methodology for simulating the distribution of incompletely similar errors. However, the proposed methodology and the steps to be followed to implement it, in further studies are not clearly and sufficiently detailed. The methodology is applied a case study, where the results are analysed, however, some steps in its application are omitted, mainly with regard to the correction of similarity error after determining the linear regressions.
In my opinion, a chapter could be added where the methodology is presented in detail as well as the steps to apply it.
More detailed comments:
Chapter 1:
Line 31: The introductory sentence could be improved; it is proposed the following:
Beam-to-column end-plate connection is an important feature for prefabricated steel structures. For many years, researchers have conducted significant experimental research on this type of joint and obtained many interesting results [1-3].
References [1-3]: it seems to me that there are many more relevant works in this area that can be referred to, and that they are even better suited to the object of this work.
I was hoping to find, in the introduction, a more detailed description of the state of the art, with regard to the existing methods of measuring and analysing errors in connection models. However, an analysis is made of some works that used models to characterize connections, without mentioning their contribution to the detection of errors, which is the focus of this work.
I believe that the introduction can be improved in that respect.
The description, at the end of the introduction, of the objectives and the applied methodology seems adequate.
Chapter 2:
Line 83: of the 16 physical quantities, 3 [F] [L] [T] are chosen, however the quantity [T] has no correspondence in the previous Table. The correspondence and meaning must be clarified.
Note the Table of the physical quantities should have a caption.
Despite referring to the basic principles associated with equations (1) to (11), it is necessary to better explain the procedure, the specific calculations as well as to identify all the variables involved and their meaning. Thus, allowing to make the article more autonomous and comprehensive.
Table 3 should be accompanied by a scheme that better clarifies the dimensions to which the designations of the models and prototypes refer to.
The description of the models in Abaqus is reasonably detailed, however I was hoping that the model could be calibrated experimentally or analytically. If not, how do you ensure that the results of the various components of the joint are valid?
I propose the comparison with numerical works, carried out on the same type of connections, and with similar numerical options that have been calibrated by experimental tests, giving the article more credibility.
See for example:
- Augusto, L. Simões, C. Rebelo, J. M. Castro, Characterization of web panel components in double-extended bolted end-plate steel joints, JCSR 116 (2016) 271 e 293, https://doi.org/10.1016/j.jcsr.2015.08.022.
Augusto H, Simões da Silva L, Rebelo C, Castro JM (2017), Cyclic behaviour characterization of web panel components in bolted end-plate steel joints, Journal of Constructional Steel Research, 133: 310-333.
- Tartaglia, M. D'Aniello, R. Landolfo. The influence of rib stiffeners on the response of extended end-plate joints. Journal of Constructional Steel Research, Volume 148, September 2018, Pages 669-690
In line 132 it is stated that the FE model was carried out similar to the experimental test, however, there is no reference to tests in the rest of the document!
Some options of the FE model should be better explained, such as the adoption of 0.4 for the coefficient of friction and how it is used in the model.
It should be noted that the dimensions in Table 4 are of the finite element mesh.
The constitutive law of the materials defined in table 5 should be described in more detail. How the stress and strain values were obtained, and whether they are true stress true strain values, as Abaqus demands.
The yield strain values for bolts and beam and columns appear to be misplaced. The ultimate strain for the high strength bolts seems to be exaggerated, please confirm.
3 - Results
On lines 158 and 159 the sentence is confused, please clarify.
Figures 3 and 4 should have the same scale in the ordinates for easy and direct comparison.
Figures should be referenced in the text before they appear.
Line 170, shouldn't it be Figures 3-7?
Line 175, identify which table you are referring to.
Line 181, refer to Figure 7 after "a completely similar model,".
Line 189, shouldn't it be Figures 7-19?
Which model does Figure 20 refer to?
Line 1896 shouldn't it be Figures 8-20?
Line 203, 204, identify which table you are referring to.
The sentence in lines 204 and 205 should be better explained, as it is an important finding.
The study is focused on the measurement of stresses at different points of the joint, however, in this type of connections it is more relevant to measure the force-displacement relations or bending moment-rotation of the components of the joint. Knowing the errors associated with these relationships seems to me more interesting from the point of view of characterizing the behaviour of the connections, instead of stresses.
In figure 21 it would be interesting for the scales of all images to be identical.
Line 234 missing a T in through.
Please clarify the sentence in line 235, it is said that the method is not suitable, but is it effective?
Have other methods been tested that may be better suited?
“Linear regression is not a perfect method for the error analysis involved in this article, but it is an effective method.”
4 – Discussion, 5 - Conclusions
It is concluded that linear regression is not the most suitable for Nc and Ep, corresponding to the component’s column web panel in shear and end-plate in bending. In fact, these are the components that most contribute to the non-linearity of the joint, and also the most important for its ductility, plastic capacity and energy dissipation. Whereas the other components remain substantially in the elastic regime. The components Nc and Ep seem more important for the characterization of the connection, and consequently for the measurement of errors. If this methodology cannot determine them accurately, it is necessary to understand whether the study is following the right path. The same is verified in the sentence present in lines 266 to 270.
Lines 267 and 279: shouldn't it be table 6?
It remained to explain, after determining the linear regressions, what is the algorithm for correcting and adjusting the errors, which led to the results in Table 6. This part is too short in the manuscript, there should be a chapter that explains this procedure in detail. A lot of time is spent in the manuscript to present the results of the numerical models and very little to explain the methods of measuring the error and its implementation, since it is the subject of the paper.
Round 2
Reviewer 2 Report
The authors introduced numerous improvements and enhanced the quality of their manuscript, however, several minor corrections are still necessary. After their introduction the manuscript will be ready for publication and additional review will not be necessary.
The clarification of the problems is now more satisfactory but the organization of the text must be still improved:
Line 81 – Table of the physics quantity does not have a number
Line 318 – The text: “ It can be seen in Figure 24 that…” should appear before Figure 24
Figures 29-31 are not mentioned in the text of the manuscript
Author Response
Please see the attachment.

This manuscript is a resubmission of an earlier submission. The following is a list of the peer review reports and author responses from that submission.
Round 1
Reviewer 1 Report
The concept of the similitude, i.e., similarity rule, has been extensively studied and applied for the fluid mechanics such as a flow-excited vibration problem. Thus, the application of the similitude theory to the end plate connection seems challenging.
In the present manuscript, however, the efficacy of the similitude theory for the end plate connection is unclear. As an example, the influence of the end plate thickness is examined through parametric analysis, in which the obvious results have been obtained. That is, the connection with a thin end plate exhibits the end plate bending whereas the connection with a thick end plate exhibits large stress appearance at the beam flange.
Although Figure 20 demonstrates similarity errors of different end plate thickness, that diagram is merely showing the influence of the end plate thickness on the connection behavior. The similitude is not principal in this manuscript.
Reviewer 2 Report
Authors present studies of errors caused by incomplete geometric similarities between tests of model and prototype beam-column end plate connections. The topic is on great importance from utilitary point of view. However, before considering its publication numerous comments and questions need to be answered:
1. References need to be expanded. The authors refer only to the articles in journals, completely ignoring monographs in the field of dimensional analysis. It is worth looking at least at books:
- Hossdorf H.: Model Analysis of Structures. Van Nostrand Reinhold 1974
- Gibbings J.C.: Dimensional Analysis. Springer-Verlag London Limited 2011
2. FEM model should be more widely described. There is no information about the type of elements used, the size of the elements. Please provide a more detailed description of boundary conditions. What was the method of the solution? Was it a simple linear static solution with force level or displacement level control?
3. Do the prototype and models use the same size of elements despite differences in dimensions? If so, have you checked the effect of element sizes on results?
4. Was the contact between the side surface of the column and the endplate simulated? If so, what was the type and what were the parameters?
5. For comparison, the relation between stress and structure deformation was used. Where exactly was the stress measured?
6. In figures 7-10 stress levels exceed 1000 MPa. What stress component is presented? What are the yield and ultimate stress for the chosen material? Is there a local yield point exceedance? If so, it is not possible to reliably compare the results with each other and look for linear relationships between them.
7. What is the stress distribution in the area of the bolted connection? Do the stress values in these areas change linearly as the thickness of the endplate changes?
8. For models with 4 mm and 6 mm thick endplates (fig. 7 and 8) there is a high deflection of endplate between bolts. What is the effect of this phenomenon on the stress level? Is this still linear deformation?
9. Regression curves presented in figure 25 and 26 cannot be considered correct. In figure 25 result for the model with 4 mm endplate should be omitted because of large error in comparison to other data points. Probably this high value of error is caused by out of plane deformation which can be observed in figure 7. Maybe the authors should consider the limit of using their predictions depending on the thickness of the endplate?
10. Authors present only results of numerical calculations for which the problem of incomplete modelling does not exist. What is the impact of endplate thickness on stress distribution in this element? Especially when deflections between bolts are high. What is the impact of endplate thickness on fatigue life?
11. How do the results obtained relate to the dimensional analysis presented in the introduction? Is it possible to determine the validity of the geometrical changes under consideration?
Notes for editing figures:
1. Legends in figures 2, 3, 7-12 are too small and unreadable.